# DNA methylation-mediated modulation of rapid desiccation tolerance acquisition and dehydration stress memory in the resurrection plant *Boea hygrometrica*

**Run-Ze Sun**[1], **Jie Liu**[2], **Yuan-Yuan Wang**[1,3], **Xin Deng**[1]*

**1** Key Laboratory of Plant Resources, Institute of Botany, Chinese Academy of Sciences, Beijing, China, **2** Facility Horticulture Laboratory of Universities in Shandong, Weifang University of Science and Technology, Shouguang, China, **3** College of Life Sciences, University of Chinese Academy of Sciences, Beijing, China

* deng@ibcas.ac.cn

**Data Availability Statement:** The RNA-seq and WGBS data have been deposited in the National Center for Biotechnology Information (NCBI) under the BioProject accession no. PRJNA665777. All

## Abstract

Pre-exposure of plants to various abiotic conditions confers improved tolerance to subsequent stress. Mild drought acclimation induces acquired rapid desiccation tolerance (RDT) in the resurrection plant *Boea hygrometrica*, but the mechanisms underlying the priming and memory processes remain unclear. In this study, we demonstrated that drought acclimation-induced RDT can be maintained for at least four weeks but was completely erased after 18 weeks based on a combination of the phenotypic and physiological parameters. Global transcriptome analysis identified several RDT-specific rapid dehydration-responsive genes related to cytokinin and phospholipid biosynthesis, nitrogen and carbon metabolism, and epidermal morphogenesis, most of which were pre-induced by drought acclimation. Comparison of whole-genome DNA methylation revealed dehydration stress-responsive hypomethylation in the CG, CHG, and CHH contexts and acclimation-induced hypermethylation in the CHH context of the *B. hygrometrica* genome, consistent with the transcriptional changes in methylation pathway genes. As expected, the global promoter and gene body methylation levels were negatively correlated with gene expression levels in both acclimated and dehydrated plants but showed no association with transcriptional divergence during the procedure. Nevertheless, the promoter methylation variations in the CG and CHG contexts were significantly associated with the differential expression of genes required for fundamental genetic processes of DNA conformation, RNA splicing, translation, and post-translational protein modification during acclimation, growth, and rapid dehydration stress response. It was also associated with the dehydration stress-induced upregulation of memory genes, including *pre-mRNA-splicing factor 38A*, *vacuolar amino acid transporter 1-like*, and *UDP-sugar pyrophosphorylase*, which may contribute directly or indirectly to the improvement of dehydration tolerance in *B. hygrometrica* plants. Altogether, our findings demonstrate the potential implications of DNA methylation in dehydration stress memory and, therefore, provide a molecular basis for enhanced dehydration tolerance in plants induced by drought acclimation.

relevant data are included within the manuscript and its Supporting Information files.

**Funding:** This work was supported by grants from the National Natural Science Foundation of China (No. 31770293 to XD and 31800212 to RZS, www.nsfc.gov.cn). The funders had no role in study design, data collection and analysis, decision to publish, or preparation of the manuscript.

**Competing interests:** The authors have declared that no competing interests exist.

## Author summary

Drought is a major adverse environmental condition affecting plant growth and productivity. Although plants can be trained to improved tolerance to the subsequent drought stress, most land plants are unable to recover from severe dehydration when the relative water content in their vegetative tissues drops below 20–30%. However, a small group of angiosperms, termed resurrection plants, can survive extreme water deficiency of their vegetative tissues to an air-dried state and recovered upon rehydration. Understanding the biochemical and molecular basis of desiccation tolerance is valuable for extending our knowledge of the maximum ability of plants to deal with extreme water loss. *Boea hygrometrica* is a well-characterized resurrection plant that can not only tolerate slow dehydration but also extend its ability to survive rapid dehydration after a priming process of slow dehydration and rehydration. The rapid desiccation tolerance in primed plants can be maintained for at least four weeks. Here, we utilized this system of drought acclimation-induced RDT acquisition, maintenance, and erasing to explore plant phenotypic, physiological, and transcriptional changes, as well as DNA methylation dynamics. The analyses of the effect of DNA methylation on gene expression and promoter methylation changes with differential gene expression revealed the putative epigenetic control of dehydration stress memory in plants.

## Introduction

Drought is one of the major adverse environmental factors affecting plant growth and productivity. More importantly, the impact of water-deficit (drought) stress on crop production is expected to increase further owing to the ongoing global climate variability and water crisis [1,2]. Understanding how plant responses to water limitation is, therefore, of utmost importance for maintaining productivity in both agricultural and natural field settings under water scarcity conditions [3].

As sessile organisms, plants have evolved intricate regulatory mechanisms at the morphological, physiological, biochemical, cellular, and molecular levels to efficiently respond to water deficit conditions [4]. Although seeds of angiosperms (flowering plants) withstand desiccation, most land plants are unable to recover from severe dehydration when the relative water contents (RWCs) in their vegetative tissues drop below 20–30% [5]. However, a small group of angiosperms, termed resurrection plants, can survive an extreme water deficiency of their vegetative tissues to an air-dried state and resume normal growth and development upon rehydration [6,7]. Vegetative tissues of desiccation-tolerant plants can avoid or resist water loss-induced mechanical and structural stresses through cell wall and plasma membrane folding, accumulating compatible solutes in vacuoles, and increasing vacuolation [8]. The integrity of both organellar and plasma membranes in desiccation-tolerant cells is protected through modifications of membrane composition, employment of efficient antioxidation mechanisms, and nonreducing-sugar-mediated stabilization during desiccation [9]. Specific drought- and desiccation-induced proteins, such as late embryogenesis abundant (LEA) proteins, early light-inducible proteins (ELIPs), small heat shock proteins (sHSPs), and antioxidative enzymes, may act directly as protectants during desiccation and rehydration or as enzymes that catalyze the synthesis of antioxidants protect against intracellular oxidative damage [9,10]. Recent advances in comparative genomics, transcriptomics, proteomics, and metabolomics of vegetative desiccation-tolerant versus desiccation-sensitive species, together with integrated multi-

omics data comparisons in desiccation-tolerant plants and tissues throughout dehydration and recovery of full metabolic competence, allowed the identification of a common and specific set of genes, proteins, and metabolites that are associated with cellular protection and recovery [6,9]. These observations are essential for understanding the biochemical and molecular basis of desiccation tolerance. Although desiccation does not often occur in the field, desiccation tolerance in resurrection plants is still valuable for extending our knowledge of the maximum ability of plants to deal with extreme water loss.

Accumulating evidence has shown that pre-exposing plants to diverse environmental stresses, considered as plant priming or acclimation, can alter subsequent responses and eventually prepare the plants to respond to future stress more quickly or actively, suggesting a form of stress memory [11]. Stress memory is mediated by the accumulation of signaling proteins, transcription factors, and metabolites, together with epigenetic alterations in DNA methylation, histone modifications, nucleosome remodeling, and expression of non-coding RNAs that are dynamically associated with transcription [12–15]. Enhancement of the stress memory through the activation of priming responses or the targeted modification of the epigenome may, thus, be a possible but largely unexplored way to improve stress resistance in agricultural crops [16]. Epigenetic memory for stress response and adaptation can be mitotically transmitted (short- and long-term memory) and meiotically inherited (transgenerational memory) in plants [17,18]. Recent studies have revealed that plants pre-exposed to multiple drought stress episodes exhibit more rapid adaptive changes in gene expression modulation during subsequent drought stress than plants experiencing drought stress for the first time [19–23]. Accordingly, genes that provide altered responses (changes in the transcription rate or transcript abundance) in subsequent stress episodes were referred to as 'memory genes', whereas those genes responding similarly to each stress episode form the 'non-memory' category [12,19,21]. Transcriptional changes in memory genes, or so-called trainable genes, are associated with the endogenous phytohormone abscisic acid (ABA) signaling pathway and epigenetic markers such as histone H3 Lys4 trimethylation (H3K4me3), DNA methylation, and long non-coding RNAs (lncRNAs) [19,24–26]. However, the detailed epigenetic mechanisms regulating transcriptional memory responses are still not fully understood.

The small perennial herbaceous plant *Boea hygrometrica* (Bunge) R. Br. (also known as *Dorcoceras hygrometricum* Bunge) is a homoiochlorophyllous resurrection species distributed widely from the tropics to the northern temperate regions of East Asia [27]. Most *B. hygrometrica* plants grown in natural habitats where water is only periodically available can survive both slow soil-drying and, the more severe, rapid air-drying stress. However, when *B. hygrometrica* plants grow under well-irrigated conditions, rapid desiccation tolerance (RDT) can be induced after a cycle of drought acclimation [27]. Previous global transcriptome and gas chromatography–mass spectrometry (GC–MS)-based metabolomics analyses have revealed the putative involvement of genes related to autophagy, ubiquitination, and ABA signal transduction, as well as metabolites such as maltose, glutaric acid, L-tryptophan, and α-tocopherol in the process of drought acclimation-induced RDT acquisition [28,29]. Recent progress in the availability of high-quality draft genome sequences and improvements in sequencing technologies have facilitated high-resolution transcriptome maps and single-base resolution methylome analyses of *B. hygrometrica* [30,31]. We have recently observed that slow drought acclimation-induced RDT in *B. hygrometrica* could be fully maintained for at least four weeks but was completely erased after 18 weeks. In this study, we utilized this system of drought acclimation-induced RDT acquisition, maintenance, and erasing in *B. hygrometrica* to explore the phenotypic, physiological, and transcriptome changes, and DNA methylation dynamics during this process, in comparison with the non-acclimated plants grown and dried in parallel. To broaden our understanding of the epigenetic control of dehydration stress memory, we

analyzed the effect of DNA methylation on gene expression and the association of promoter methylation changes with differential gene expression.

## Results

### Phenotypic and physiological responses of *B. hygrometrica* during RDT acquisition, maintenance, and erasing

*B. hygrometrica* seedlings grown under well-watered, greenhouse conditions were able to survive slow soil-drying (soil-dried plants, SD) and acquire tolerance to rapid desiccation (air-dried acclimated plants, AD) after rehydration (acclimated plants, A) (Fig 1A). Such drought acclimation-induced RDT in plants could be maintained for 4 to 13 w (A4, A8, and A13) but

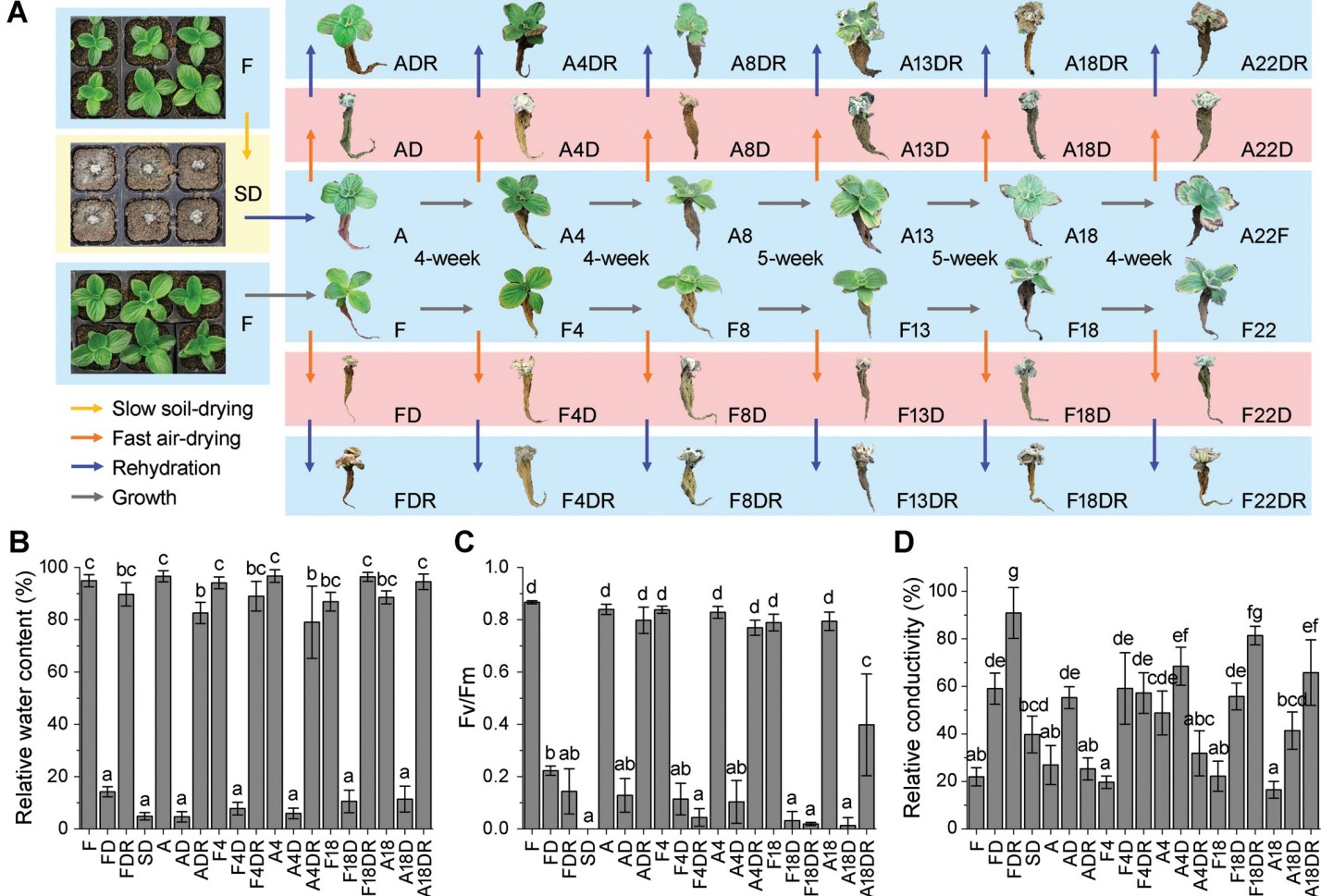

**Fig 1. Physiological analysis of dehydration stress treated *B. hygrometrica* plants.** (A) Diagram of the RDT acquisition, maintenance, and erasing processes in *B. hygrometrica* plants. Three-month-old fresh (F) plants of *B. hygrometrica* failed to resurrect after rapid air-drying (FD) and subsequent rehydration (FDR). After slow soil-drying for 2 w (SD) and rehydration for 3 d, the fresh plants became acclimated (A) and were able to resurrect after rapid air-drying (AD) and subsequent rehydration (ADR). Such acclimation-primed RDT can be maintained in plants after continuous cultured for 4 w (A4, A4D, and A4DR) to 8 and 13 w (A8, A8D, and A8DR; A13, A13D, and A13DR), but the ability of RDT was lost after culture for 18 and 22 w (A18, A18D, and A18DR; A22, A22D, and A22DR). Their corresponding control groups (F, F4, F8, F13, F18, and F22), always unable to resurrect after rapid air-drying (FD, F4D, F8D, F13D, F18D, and F22D) and subsequent rehydration (FDR, F4DR, F8DR, F13DR, F18DR, and F22DR). (B-D) Determination of leaf RWC (B), *Fv/Fm* (C), and REC (D) in *B. hygrometrica* plants during the rapid desiccation tolerance acquisition, maintenance, and erasing processes. Data are presented as the mean ± SD of four to six biological replicates (each replicate with 6 plants), and different superscript letters over bars represent statistical differences at the 0.01 level.

was completely erased 18 weeks after acclimation (A18 and A22) (Fig 1A). In contrast, the non-acclimated fresh plants (F, F4, F8, F13, F18, and F22) failed to resurrect after rapid air-drying (FD, F4D, F8D, F13D, F18D, and F22D) and subsequent rehydration (FDR, F4DR, F8DR, F13DR, F18DR, and F22DR) during the entire growth stage (Fig 1A). The leaf RWC of *B. hygrometrica* decreased to a similar level (< 20%) in all samples subjected to either slow soil-drying for 14 d or rapid air-drying for 2 d; also, leaves presented a significant decrease ($p < 0.01$) in photosynthetic efficiency based on the value of *Fv*/*Fm* (Fig 1B and 1C). However, the photosynthetic efficiency of dehydrated leaves from neither non-acclimated control groups (FD, F4D, and F18D) nor RDT-erased acclimated plants (A18D) restored to normal levels after rehydration (FDR, F4DR, F18DR, and A18DR), although their leaf RWC was rapidly recovered to approximately 80%. Moreover, a significant increase ($p < 0.01$) in leaf relative electrical conductivity (REC) was observed in all fresh plants subjected to dehydration stress (SD/F, FD/F, AD/A, A4D/A4, F4D/F4, A18D/A18, and F18D/F18), but it could be restored only in the plants that acquired or maintained RDT (A/SD, ADR/AD, and A4DR/A4D) (Fig 1D). These data provide additional evidence for the time-course of acquiring, maintaining, and erasing drought acclimation-induced RDT in *B. hygrometrica* plants.

## Global transcriptome changes in *B. hygrometrica* during RDT acquisition, maintenance, and erasing

To elucidate the molecular basis for the acquisition, maintenance, and erasing of the drought acclimation-induced RDT in *B. hygrometrica*, we characterized the dehydration-induced transcriptome dynamics in leaves of both acclimated and non-acclimated plants from different growth stages under normal conditions with three biological replicates. Approximately 6.58 Gb of clean bases (clean reads Q20 > 98%) were obtained from each sample by high-throughput RNA sequencing (RNA-seq), with an average unique genome mapping ratio of 68.24% (see details in S1 Table). A total of 23,892 novel transcripts containing 19,449 putative protein-coding transcripts were reconstructed, whereas a significant number of transcripts representing 2,239 potential novel genes not yet identified in the reference genome. An initial examination of the overall transcriptome data using multivariate statistics including hierarchical clustering analysis (HCA) and principal component analysis (PCA) revealed that most of the individual replicates clustered closely (S1 Fig), indicating relatively low biological variability among triplicate samples. There was a rough separation in the transcriptome of fresh plants and plants suffering from slow and rapid dehydration, as evidenced by the detection of large numbers of differentially expressed genes (DEGs) between fresh and dehydrated plants (Fig 2A). The highest oscillations in global gene expression occurred during the drought acclimation process (SD/F and A/SD) and the rapid dehydration process of plants at early growth stages (AD/A, FD/F, A4D/A4, and F4D/F4) (Fig 2A). Large-scale reprogramming of the transcriptome was also observed in both acclimated and non-acclimated plants after continuous culture for 18 w (A18/A and F18/F) and subsequent rapid dehydration process (A18D/A18 and F18D/F18) (Fig 2A). In contrast, relatively small numbers of DEGs were detected between acclimated and non-acclimated fresh plants during early growth stages (A/F, A4/F4, A4/A, and F4/F) and dehydrated plants during later growth stages (A4D/F4D and A18D/F18D) (Fig 2A). Therefore, it may be suggested that drastic alternations in the transcriptional activity of genes occur during long-term growth and the dehydration stress response in young *B. hygrometrica* seedlings.

## Identification of putative RDT-related memory genes

Comparisons of the list of up- and down-regulated DEGs between acclimated and non-acclimated fresh plants at three growth stages were performed to identify candidate genes

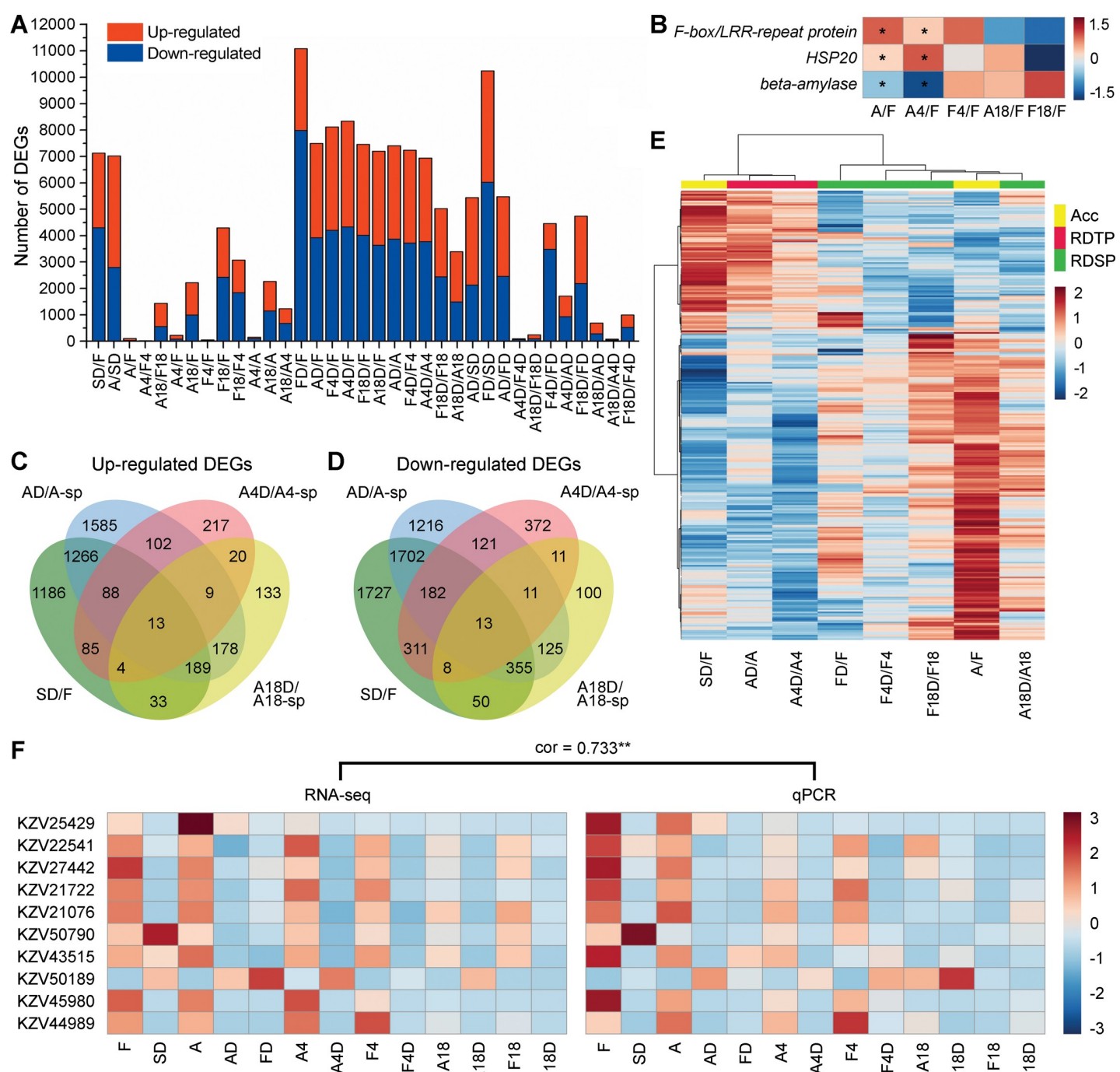

**Fig 2. Transcriptome changes in *B. hygrometrica* plants during acclimation and responding to rapid dehydration stress.** (A) Number statistic of DEGs between samples. (B) Expression changes of acclimation-induced and -inhibited genes in acclimated/non-acclimated fresh plants during dehydration stress memory. Scale bar represents log2 fold-change expression (red, upregulation; blue, downregulation) between samples. Genes with significant expression differences between samples (fold change $\geq$ 2 and adjusted $p \leq 0.05$) are indicated by asterisks (*) in the squares. (C and D) Venn diagrams displaying the overlap of up- (C), and down-regulated (D) DEGs specific response in fresh plants responding to slow soil-drying (SD/F) and drought-acclimated plants responding to rapid dehydration (AD/A-sp, A4D/A4-sp, and A18D/A18-sp). (E) Hierarchical clustering heatmap showing the expression changes of RDT-specific dehydration stress memory genes in plants during acclimation (SD/F and A/F) and both acclimated and non-acclimated fresh plants responding to rapid dehydration stress (FD/F, AD/A, F4D/F4, A4D/A4, F18D/F18, and A18D/A18). Acc, the acclimation process; RDTP, rapid dehydration-tolerant plants; RDSP, rapid dehydration-sensitive plants. Scale bar represents log2 fold-change expression (red, upregulation; blue, downregulation) between samples. (F) Validation of the RNA-seq data by qPCR. The average FPKM values of the RNA-seq results and the average relative expression values from three independent qPCR experiments are shown in the left- and right-hand heatmaps, respectively. Correlation is significant at the 0.01 level. Scale bar represents the normalized expression of each gene between samples.

responsible for the drought acclimation-induced RDT in *B. hygrometrica*. The analysis showed that none of the up- or down-regulated DEGs between A and F were differentially expressed between A4 and F4 (S2 Fig). Nevertheless, two genes encoding F-box/LRR-repeat protein (FBXL, KZV28834) and heat shock protein 20 (HSP20, KZV34970) and the gene encoding *β*-amylase (BAM, KZV36643) were up- and down-regulated (fold change ≥ 2, adjusted $p ≤ 0.05$) in rapid desiccation-tolerant plants (A and A4) compared with the initial control-group plants (F), respectively (Figs 2B and S3). In contrast, there were no significant differences in their transcript abundances between rapid desiccation-sensitive (F4, F18, and A18) and the initial control-group plants (Fig 2B).

Meanwhile, we compared the list of rapid dehydration-induced up- and down-regulated DEGs between the drought-acclimated plants (AD/A, A4D/A4, and A18D/A18) and their respective non-acclimated control groups (FD/F, F4D/F4, and F18D/F18), respectively (S4 Fig). In total, 493 DEGs, including 190 up- and 303 down-regulated genes, were identified as specific rapid dehydration-responsive genes in rapid desiccation-tolerant plants (AD/A and A4D/A4). These genes showed no significant transcriptional changes or opposite responses of RDT-erased plants to rapid dehydration stress (A18D/A18) and their corresponding control groups (FD/F, F4D/F4, and F18D/F18) (Figs 2C and 2D, and S5). Functional categories of the RDT-specific DEGs were classified according to the Gene Ontology (GO) term and Kyoto Encyclopedia of Genes and Genomes (KEGG) pathway enrichment analyses. GO term enrichment analysis revealed that the RDT-specific DEGs were enriched in the biological processes of auxin polar transport, lipid metabolism, plant epidermal cell differentiation, tissue development, and negative regulation of post-embryonic development, but this was not statistically significant ($p ≤ 0.05$, Q-value > 0.05) (S2 Table). The KEGG pathway enrichment analysis showed that histidine metabolism, zeatin biosynthesis, nicotinate and nicotinamide metabolism, and *α*-linolenic acid metabolism were enriched ($p ≤ 0.05$, Q-value > 0.05) by these RDT-specific DEGs in rapid desiccation-tolerant plants (S2 Table). Remarkably, approximately 46% (88 genes) and 60% (182 genes) of the RDT-specific up- and down-regulated DEGs, respectively, showed the same transcriptional changes in fresh plants subjected to slow soil-drying (Fig 2C and 2D), thus representing putative mid-term dehydration stress memory genes that were acquired after drought acclimation and maintained during early growth stage (A and A4) in *B. hygrometrica*. Almost all of these RDT-specific dehydration stress memory genes that were pre-induced after slow soil-drying (SD/F) decreased their transcription back to basal levels after rehydration (A/F) (Fig 2E). In addition, a total of 6,616 DEGs, including 3,218 up- and 3,398 down-regulated genes, were identified as specific rapid dehydration-responsive genes in drought-acclimated plants at the initial growth stage (AD/A), in comparison with those at later growth stages (Fig 2C and 2D). Among them, approximately 45% (1,455 genes) and 61% (2,057 genes), defined as putative short-term dehydration stress memory genes, were pre-induced by drought acclimation but erased at the later growth stage (A4D/A4) and showed no significant transcriptional changes or opposite responses in non-acclimated plants (FD/F and F4D/F4) (Figs 2C and 2D, and S6). Moreover, a total of 48 DEGs, including 22 up- and 24 down-regulated genes, were identified as specific rapid dehydration-responsive genes in drought-acclimated plants during the entire growth stage (AD/A, A4D/A4, and A18D/A18) (Fig 2C and 2D), and more than half of these genes were pre-induced by drought acclimation (defined as putative long-term dehydration stress memory genes) (S7 Fig). These data suggest that drought acclimation may contribute to a more rapid and robust response of gene expression changes under rapid dehydration stress.

To validate the RNA-seq data, the relative expression levels of 10 genes randomly selected from the lists of up- and down-regulated RDT-specific DEGs were determined in each sample by quantitative real-time PCR (qPCR). The transcription profiles of these genes in both

acclimated and non-acclimated plants during growth and dehydration stress obtained from an independent qPCR evaluation was significantly positively correlated [Pearson correlation coefficient (cor) = 0.733, $p < 0.001$] with those from the transcriptomes (Fig 2F), indicating the high reproducibility of the RNA-Seq results.

## Transcriptional profiles of methylation pathway genes in *B. hygrometrica* during RDT acquisition, maintenance, and erasing

To investigate whether DNA methylation is involved in the process of drought acclimation-induced RDT acquisition, maintenance, and erasing in *B. hygrometrica* plants, we inspected genes encoding DNA methylation and demethylation-related enzymes at the transcriptomic scale; we also assayed their expression profile changes in both acclimated and non-acclimated plants during growth and dehydration stress. Several DNA methylation pathway genes were identified from the *B. hygrometrica* transcriptome; 10 genes encoding DNA methyltransferase 1/chromomethyltransferase 3 (MET1/CMT3), 9 genes encoding DNA (cytosine-5)-methyltransferase (DNMT), 8 genes encoding histone deacetylase (HDAC), 1 gene encoding decrease in DNA methylation 1 (DDM1), 2 genes encoding RNA-directed DNA methylation 4 (RDM4), 1 gene encoding factor of DNA methylation 1 (FDM1), and 5 genes encoding transcriptional activator DEMETER-like DNA demethylase/repressor of silencing 1 (DML/ROS1) (Fig 3A). No significant difference was detected in the expression of genes encoding methylation or demethylation-related enzymes between the acclimated and non-acclimated fresh plants at early growth stages (A/F and A4/F4); only two genes encoding HDAC (KZV52430) and DNMT (KZV26154) involved in the maintenance of CG methylation and a gene encoding DML (KZV31242) required for DNA demethylation were presented significantly higher and lower expression, respectively, in drought-acclimated fresh plants after continuous culture for 18 w than those of the non-acclimated plants (A18/F18) (Fig 3A). In contrast, members of several enzyme families, including HDA, DNMT, and MET1/CMT3, required for the establishment and maintenance of DNA methylation, exhibited significant dehydration-induced transcriptional changes in acclimated or non-acclimated plants (Fig 3A). Nevertheless, the transcription of a gene encoding FDM1 (KZV30079), involved in RNA-directed DNA methylation (RdDM), and several members of the DML enzyme family were significantly or moderately down-regulated in both acclimated and non-acclimated fresh plants in response to dehydration stress (Fig 3A). Furthermore, among these DNA methylation pathway genes, it is obvious that a higher number of genes were down-regulated than those that were up-regulated in response to dehydration stress (Fig 3A). Taken together, these observations indicate that DNA methylation pathways are functionally conserved and may play an active role in the acquisition, maintenance, and erasing of RDT in *B. hygrometrica*.

## Dynamic DNA methylation landscapes in *B. hygrometrica* during RDT acquisition, maintenance, and erasing

The acclimation-dependent induction of rapid dehydration stress-responsive genes (memory genes) may be associated with acclimation-related and dehydration-induced changes in DNA methylation, which play an important role in response to drought stress. To test this, we employed genomic DNA methylation profiling at single-base resolution to survey genome-wide DNA methylation variations in the leaves of both acclimated and non-acclimated *B. hygrometrica* plants during growth and dehydration stress. High-throughput whole-genome bisulfite sequencing (WGBS) yielded an average of 51.01 Gb clean bases (Q30 > 83.24%) from each sample, and approximately 84.35% of reads were mapped to the reference genome, corresponding to an effective depth of 28.96-fold coverage (see details in S3 Table). Global DNA

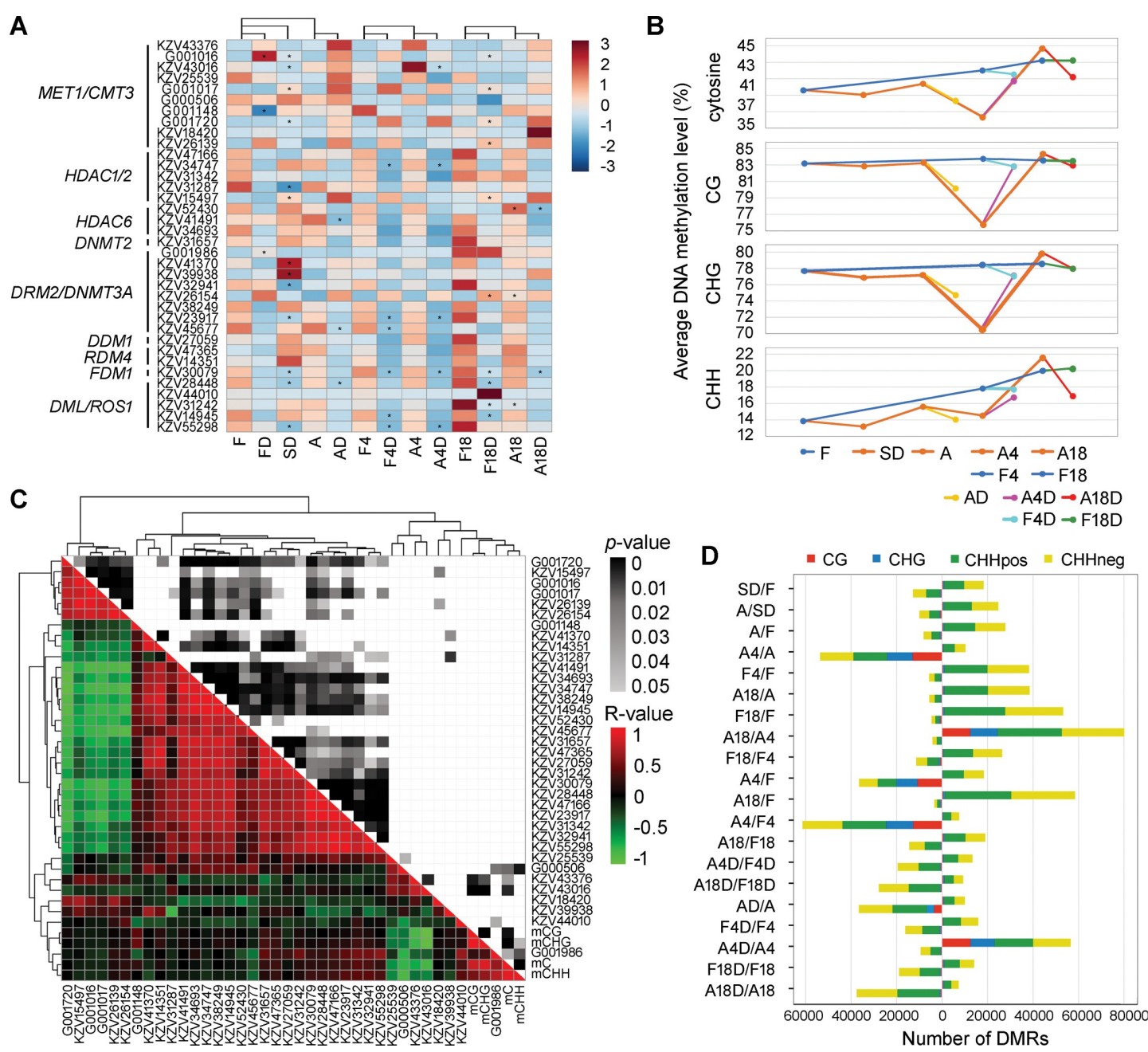

**Fig 3. DNA methylation dynamics in *B. hygrometrica* plants during acclimation and responding to rapid dehydration stress.** (A) Expression profile of genes encoding DNA methylation and demethylation-related enzymes during drought acclimation, growth, and response to dehydration stress. MET1/CMT3, DNA methyltransferase 1/chromomethylase 3; HDAC, histone deacetylase; DNMT, DNA methyltransferase; DRM2, domains rearranged methyltransferase 2; DDM1, decrease in DNA methylation 1; RDM4, RNA-directed DNA methylation 4; FDM1, factor of DNA methylation 1; DML/ROS1, DEMETER-like DNA demethylase/repressor of silencing 1. Scale bar represents the normalized expression of each gene between samples. Genes with significant expression changes (fold change ≥ 2 and adjusted $p ≤ 0.05$) in drought-acclimated plants compared with the non-acclimated control groups (A/F, A4/F4, and A18/F18) and in plants subjected to dehydration stress (SD/F, FD/F, A/F, F4D/F4, A4D/A4, F18D/F18, and A18D/A18) are indicated by asterisks (*) in the corresponding squares. (B) Global changes in the average proportion of methylated C, CG, CHH, and CHG. (C) Correlation analysis between the level of DNA methylation and the expression of methylation pathway genes. (D) Number statistic of DMRs between samples. Hypo- and hyper-DMRs are shown in the left and right columns, respectively.

methylation profiles showed that the methylcytosines (mCs) occupied 36.35–44.67% of all cytosine sites in the genome of *B. hygrometrica* plants at different growth stages, with or without dehydration stress (Fig 3B). The average methylation level was higher in CG and CHG

(H = A, T, or C) contexts (75.75–84.32% and 70.48–79.76%, respectively), in comparison with that in the CHH context (13.20–20.20%) within each sample (Fig 3B). Global genomic mC frequency and the average methylation level in CHH context gradually increased in non-acclimated plants with extended periods of culturing but were maintained at lower levels in plants after SD and rapid desiccation-tolerant plants (A and A4), sharply increasing when the memory of RDT was erased (A18) (Fig 3B). The frequency of mC in each of the three contexts was moderately decreased in all fresh plants subjected to dehydration stress, except for a dramatic increase in the comparison of A4D/A4 mainly because of the relatively low methylation levels of mCG and mCHG in A4 (Fig 3B). The average DNA methylation levels of each cytosine context in plants after slow soil-drying and both fresh and dehydrated rapid desiccation-tolerant plants (A, AD, A4, and A4D) were lower than those in rapid desiccation-sensitive plants (F, F4, F4D, F18, F18D, A18, and A18D) (Fig 3B), indicating that DNA methylation might be associated with the process of RDT acquisition, maintenance, and erasing. In this context, correlation analysis based on Pearson's coefficient revealed that the transcription of a novel gene encoding DNMT3A (G001986) was significantly positively correlated ($p \leq 0.05$) with the average genomic frequency of mC and mCHH among samples (Fig 3C). Moreover, significant negative correlations were observed between changes in the transcription of genes encoding MET1/CMT3 (KZV43016 and KZV43376) and average genomic frequency of mC, mCG, and mCHG, as well as between changes in the transcription of a novel MET1/CMT3 family member (G000506) and average genomic frequency of mC and mCHH (Fig 3C).

To further study the DNA methylation dynamics in the processes of RDT acquisition, maintenance, and erasing, we identified the differentially methylated regions (DMRs) between samples. Local changes in DNA methylation were more frequent in the CHH context than in the CG and CHG contexts between acclimated and non-acclimated plants and in plants responding to dehydration stress at each growth stage (Fig 3D). Consistent with the changes in average mCG, mCHG, and mCHH levels in *B. hygrometrica* during the process of drought acclimation, similar numbers of hyper- and hypo-DMRs in the CG and CHG contexts were detected in the comparisons of SD/F and A/SD, respectively, but the number of hype-DMRs in the CHH context was greater than that of hypo-DMRs in the CHH context in the comparison of A/SD (Fig 3B and 3D). Higher numbers of hypo-DMRs in the CG, CHG, and CHH contexts were found in the comparisons of A4/A, A4/F, and A4/F4, while the number of hyper-DMRs in all three cytosine contexts was greater in the comparisons of A18/A4 and A4D/A4 (Fig 3D), indicating a potential relationship between DNA hypomethylation and the maintenance of RDT. In addition, most DMRs were hypo-DMRs in the CHH context in the comparisons of AD/A and A18D/A18 but were hyper-DMRs in the CHH context in the comparisons of F4/F, F18/F4, F18/F, A18/A, and A18/A4, which were consistent with the changes in average mCHH levels in fresh plants during different growth stages (Fig 3B and 3D).

## Effect of DNA methylation on gene expression

DNA methylation plays an essential role in the regulation of gene expression in plants. To explore the association between DNA methylation and gene activity, we compared the methylation levels of CG, CHG, and CHH contexts within the gene body (exon and intron) and their flanking regions (2-kb sequences upstream and downstream of the protein-coding regions, respectively) among the genes with low (FPKM $\leq 1$), medium ($1 <$ FPKM $\leq 100$), and high (FPKM $> 100$) transcript abundances. As shown in Fig 4A, increased CG methylation in gene bodies and flanking regions and reduced methylation at both transcription start sites and transcription end sites were detected in each sample. Meanwhile, both the CHG and CHH methylations were decreased within gene bodies (especially in their intron regions) but elevated in

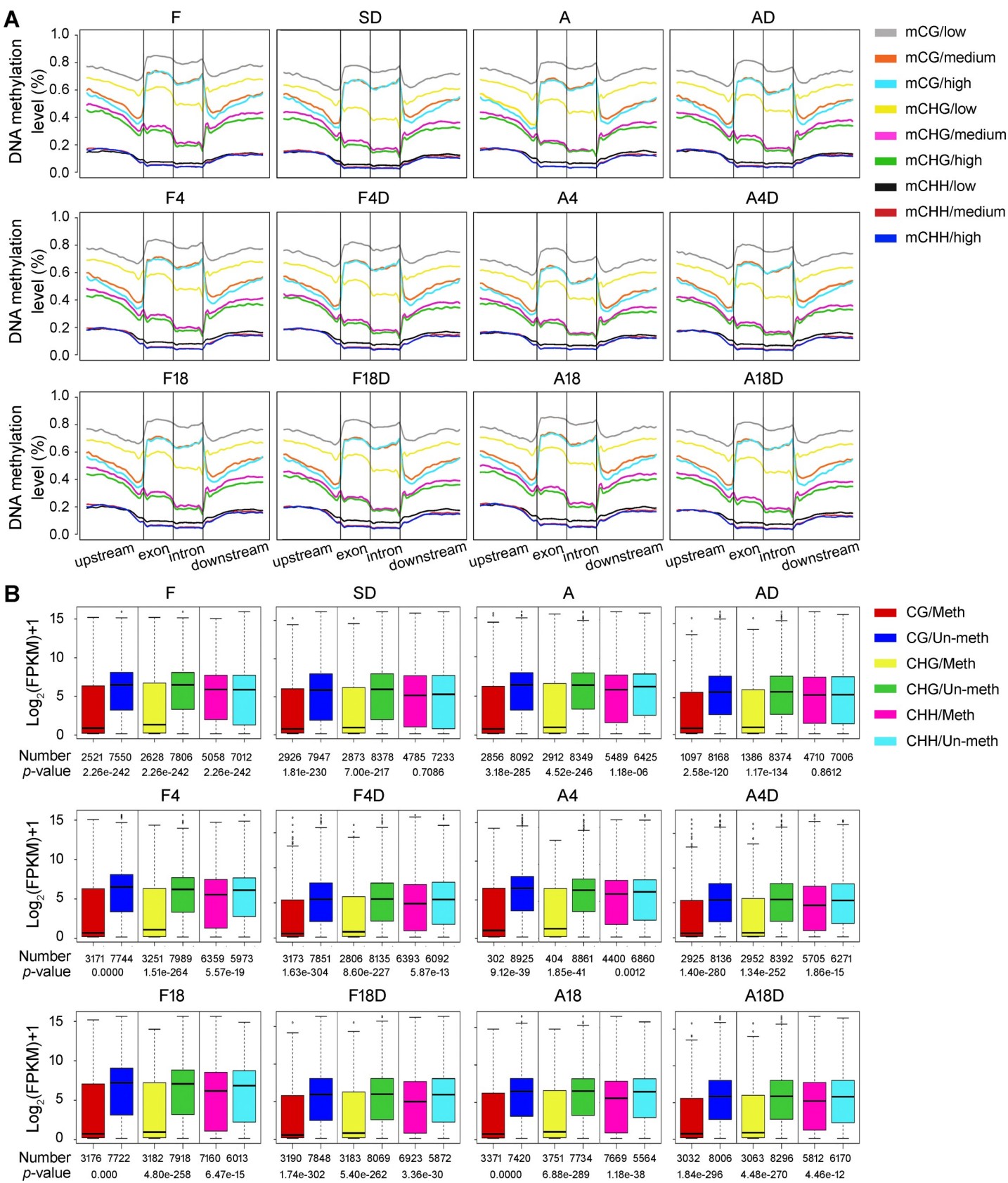

**Fig 4. Effect of DNA methylation on gene expression in each sample.** (A) Relationship between the transcript abundance (low, medium, and high) of genes and the level of DNA methylation in the CG, CHG, and CHH contexts within the gene body (exon and intron) and flanking regions (2-kb sequences upstream and downstream of the protein-coding region, respectively). (B) Comparison of the transcript abundance between promoter-methylated and -unmethylated genes. Numbers of promoter-methylated and unmethylated genes, as well as the Wilcoxon rank-sum test *p*-value of each sample, are indicated below the diagram. Meth, promoter-methylated genes; Un-meth, promoter-unmethylated genes.

their upstream and downstream flanking regions (Fig 4A). The average level of DNA methylation in the CG and CHG contexts within the gene body and flanking regions and in the CHH context within the gene body and downstream flanking regions were much higher in genes with low transcript abundances than those with medium and high transcript abundances within most individual samples (Fig 4A). Moderate differences in average methylation levels in the CG context within the flanking regions and CHG context within the gene body and flanking regions were also observed between the genes with medium and high transcript abundances in each sample (Fig 4A). However, the global methylation levels of CG, CHG, and CHH contexts within both the gene body and flanking regions showed no obvious correlation with up- or down-regulation of gene expression in different growth stages or dehydration stress, although there were only slight differences in the methylation levels of mCHG and mCG contexts between up- and down-regulated genes in the comparisons of A/F, F4/F, and A4D/F4D (S8 Fig). These observations suggest important functions of both CG and non-CG methylation in genic and intergenic regions in gene silencing within individual samples; however, this methylation may play a limited role in determining differential gene expression between samples.

It is well established that DNA methylation at promoter regions inhibits gene transcription. In the present study, we calculated the average methylation levels of cytosines in the CG, CHG, and CHH contexts within the 2-kb sequences upstream of the protein-coding regions (considered as promoter regions) of all genes (S4 Table), and defined the genes with high (above the upper quartile) and low (below the lower quartile) methylation levels in the promoter regions as promoter-methylated and promoter-unmethylated genes, respectively. Comparison of the transcript abundance between promoter-methylated and promoter-unmethylated genes revealed a highly significant negative correlation ($p < 0.001$) between gene expression levels and promoter methylation levels, especially in the CG and CHG contexts (Fig 4B). This analysis implicates the involvement of DNA methylation/demethylation at promoter regions in the silencing/activation of gene expression in *B. hygrometrica* plants, irrespective of their growth stages or stress conditions.

## Association of promoter methylation changes with differential gene expression

To further investigate the effect of promoter methylation on transcriptional changes in *B. hygrometrica* during RDT acquisition, maintenance, and erasing progress, we compared the fold-change of gene expression (log2 ratio) with the methylation differences in the three cytosine contexts within their promoter regions between samples. Pearson correlation analysis showed no significant correlation ($p > 0.05$) between gene expression changes and DNA methylation variations within their promoter regions during acclimation, growth, or dehydration stress (Figs 5 and S9). Similarly, no significant relationship ($p > 0.05$) was found between the transcriptional changes of genes with different methylation levels in their promoter regions and the corresponding promoter methylation variations (S10 Fig). The up- and down-regulated DEGs with significantly decreased and increased methylation levels, respectively, within either or both of the two symmetric cytosine contexts between samples, nevertheless, could

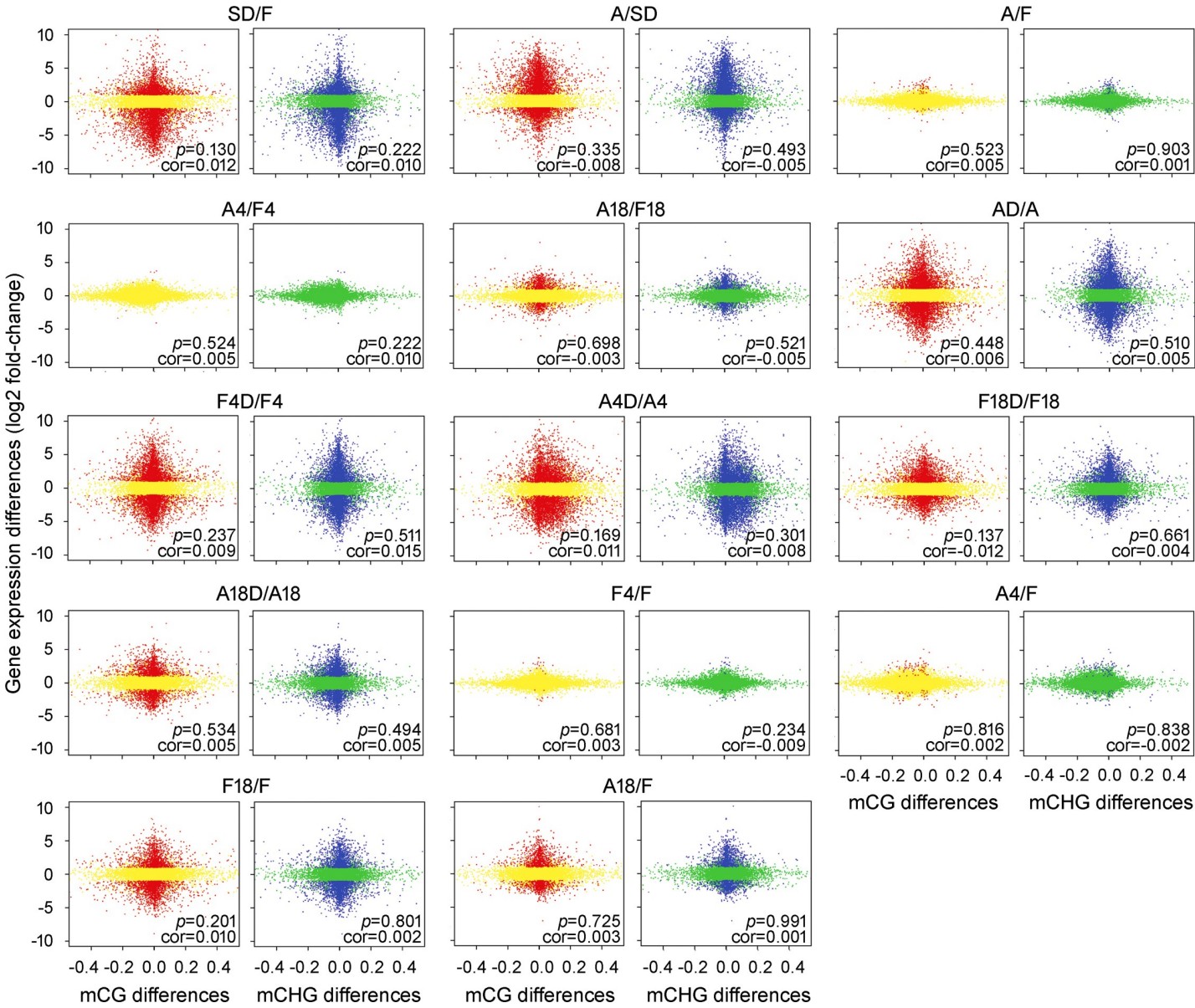

**Fig 5. Correlation analysis of gene expression differences and promoter methylation variations in the CG and CHG contexts between samples.** DEGs and non-DEGs in the differential mCG plots are indicated by red and yellow dots, respectively. DEGs and non-DEGs in the differential mCHG plots are indicated by blue and green dots, respectively. Cor, Pearson correlation coefficient.

still be considered as candidate methylation-regulated DEGs (MRGs) potentially controlled by DNA methylation. On one hand, a total of 109 and 102 putative MRGs were screened from genes exhibiting opposite changes in promoter methylation and transcription levels during slow soil-drying and subsequent rehydration, respectively (Figs 5 and S11). GO enrichment analysis revealed that these acclimation-related MRGs were insignificantly enriched ($p \leq 0.05$, Q-value > 0.05) in the biological processes of actin cytoskeleton organization and the biosynthetic processes of extracellular polysaccharides and inosine-5′-monophosphate (IMP) (Fig 6A and S5 Table). On the other hand, a total of 27, 153, and 1,278 putative MRGs were screened out from DEGs between acclimated and non-acclimated fresh plants (A/F, A4/F4, and A18/F18), and during plant growth (A/F, A4/F, A18/F, F4/F, and F18/F), and responding to rapid

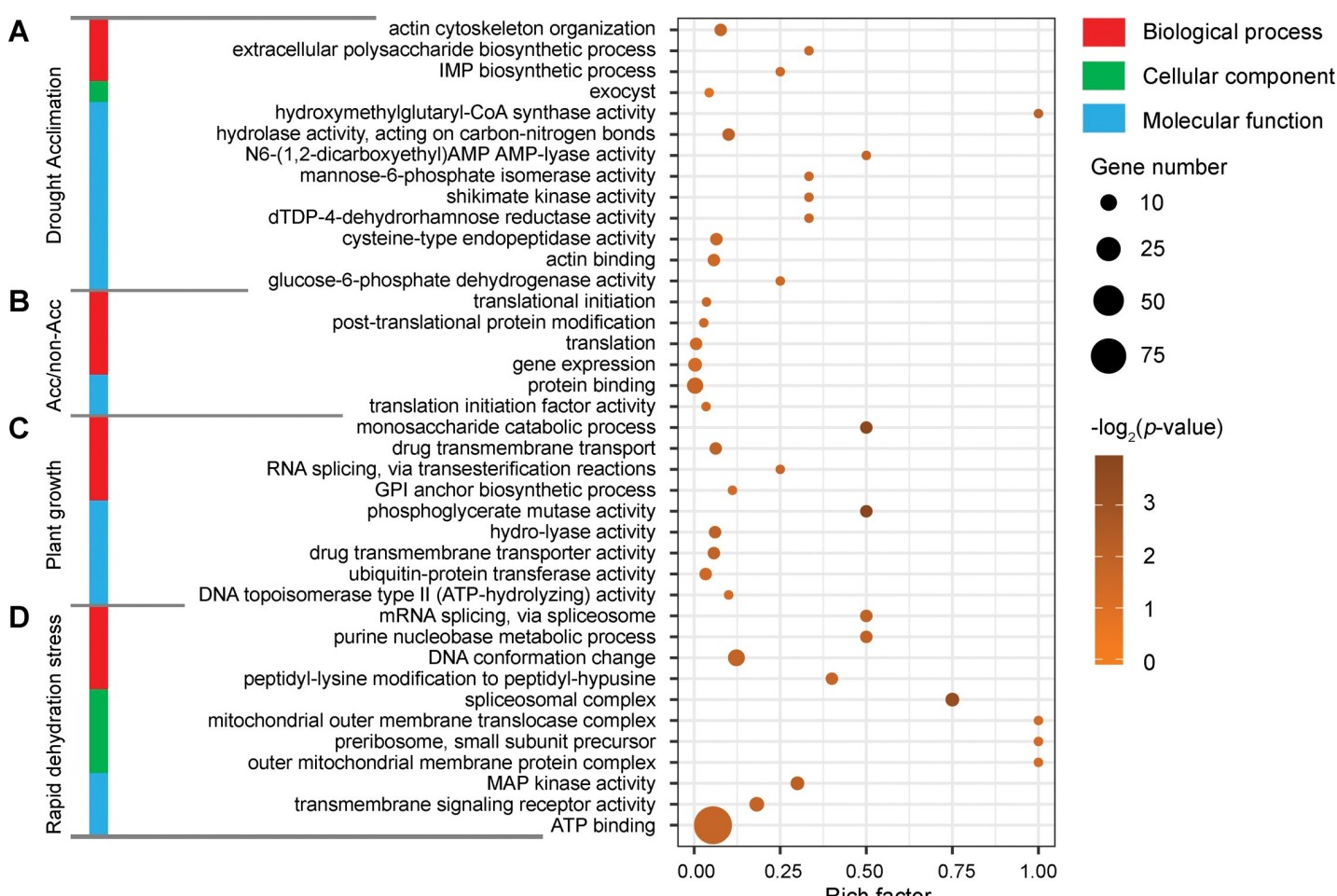

**Fig 6.** Gene Ontology enrichment analyses of the putative MRGs during drought acclimation (A), between acclimated and non-acclimated fresh plants (B), and during plant growth (C) and rapid dehydration stress response (D).

dehydration stress (AD/A, A4D/A4, F4D/F4, A18D/A18, and F18D/F18), respectively (S11 Fig). GO analysis revealed a statistically insignificant enrichment ($p \leq 0.05$, Q-value > 0.05) of biological processes associated with translation and post-translational protein modification and molecular functions associated with protein binding and translation initiation factor activity among the 27 MRGs between acclimated and non-acclimated fresh plants (Fig 6B and S6 Table). Significant enrichment ($p \leq 0.05$, Q-value $\leq 0.05$) of the biological process associated with monosaccharide catabolism and the molecular function of phosphoglycerate mutase activity was recorded based on the MRGs during growth (Fig 6C and S7 Table). Several other biological processes, including drug transmembrane transport, RNA splicing, and glycosyl-phosphatidylinositol anchor biosynthesis, and molecular functions, including hydro-lyase, drug transmembrane transporter, ubiquitin-protein transferase, and DNA topoisomerase II activities, were also enriched by MRGs during growth, but without statistical significance ($p \leq 0.05$, Q-value > 0.05) (Fig 6C and S7 Table). In addition, the rapid dehydration stress-responsive MRGs were insignificantly enriched ($p \leq 0.05$, Q-value > 0.05) in the GO biological processes of mRNA splicing, purine nucleobase metabolism, DNA conformation change, peptidyl-lysine modification to peptidyl-hypusine, cellular components of preribosomes, spliceosomal, mitochondrial outer membrane translocase, and outer mitochondrial membrane

protein complexes, as well as molecular functions of MAP kinase activity, transmembrane signaling receptor activity, and ATP binding (Fig 6D and S8 Table).

## Characterization of putative methylation-regulated dehydration stress memory genes

To evaluate the impact of dehydration stress-induced DNA methylation alternations on transcriptional memory responses, we identified subsets of genes between the lists of putative MRGs and dehydration stress memory genes. Intriguingly, a number of acclimation-related and/or dehydration stress-responsive MRGs were found among the lists of short-term (175 genes) and mid-term (14 genes) dehydration stress memory genes (Fig 7A and 7B and S9 Table), but were absent from the list of long-term dehydration stress memory genes. GO enrichment analysis revealed that the putative methylation-regulated dehydration stress memory genes were insignificantly enriched ($p \leq 0.05$, Q-value $> 0.05$) in the biological processes of proline catabolic process, negative regulation of microtubule depolymerization, NADH dehydrogenase complex (plastoquinone) assembly, urea transport, cellular components of spindle microtubule, photosystem, cell cortex, intrinsic component of membrane, and molecular functions of FMN binding, microtubule plus-end binding, and activities of phosphoribosylformylglycinamidine synthase, proline dehydrogenase, urea transmembrane transporter, transcription corepressor, (S)-2-hydroxy-acid oxidase, and biotin carboxylase (Fig 7C and S10 Table). Among the putative methylation-regulated mid-term dehydration stress memory genes we predicted, 11 of them displayed significantly down-regulated transcript abundances in the comparisons of SD/F, AD/A, and A4D/A4, whereas only three genes encoding pre-mRNA-splicing factor 38A (PSF38A, KZV19471), vacuolar amino acid transporter 1-like (AVT1L, KZV50189), and UDP-sugar pyrophosphorylase (USPase, KZV45485) showed the opposite pattern in the three aforementioned comparisons (S12 Fig and S9 Table). This finding suggests that DNA methylation participates in the dehydration response of RDT-specific stress memory genes, mainly through its transcriptional repression activity associated with elevated methylation levels.

To elucidate whether promoter methylation variations could be correlated with the transcriptional activity of memory genes, we determined the expression profile of the putative methylation-regulated mid-term dehydration stress memory genes in *B. hygrometrica* seedlings exposed to different concentrations of the DNA methylation inhibitor 5-azacytidine (5-azaC) and the methyl donor S-adenosyl methionine (SAM). The qPCR results showed that the relative transcription levels of most putative methylation-regulated mid-term dehydration stress memory genes were moderately ($p > 0.05$) up-regulated in plants subjected to 5-azaC treatment, and significantly ($p \leq 0.05$) or moderately down-regulated in plants subjected to SAM treatment (Fig 7D). To validate SAM-induced methylation changes, the promoter regions of two methylation-regulated mid-term dehydration stress memory genes encoding indole-3-acetic acid inducible 14 (IAA14, KZV17830) and ureidoglycolate hydrolase (UAH, KZV21076) were subjected to bisulfite sequencing PCR (BSP). The results revealed that both genes showed an increase in DNA methylation within their promoter regions after SAM (100 μM) treatment (Fig 7E), further demonstrating the putative role of DNA methylation at promoter regions in the silencing of gene expression.

## Discussion

Drought stress has become a major concern for agricultural production worldwide because it greatly affects the growth, development, and productivity of plants. The main effects of drought stress caused by atmospheric, soil, or physiological drought are decreases in leaf water

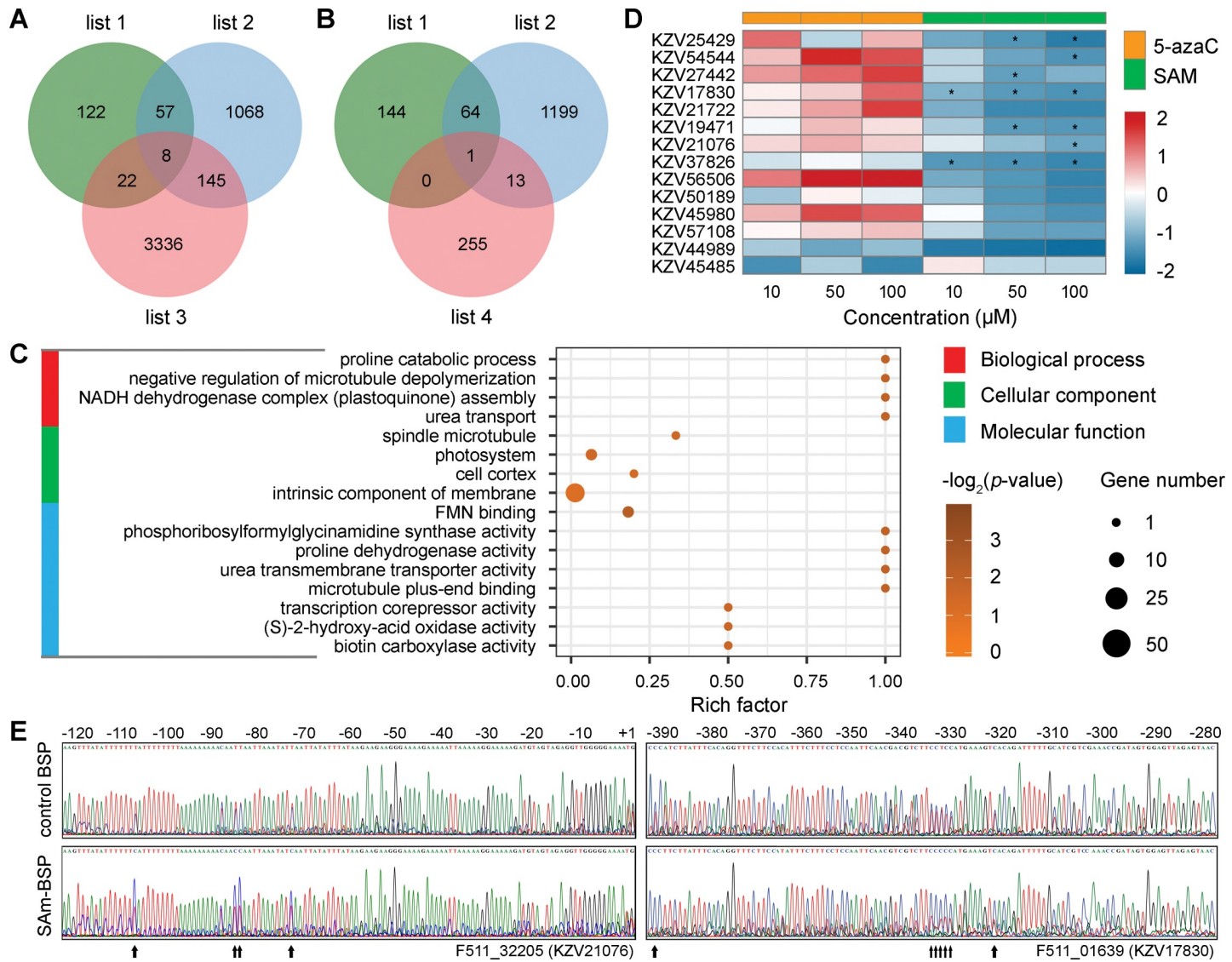

**Fig 7. Screening and validation of the putative methylation-regulated dehydration stress memory genes.** (A and B) Venn diagrams displaying the overlap of acclimation-related (list 1) and rapid dehydration stress-responsive (list 2) MRGs with short-term (list 3 in A) or mid-term (list 4 in B) dehydration stress memory genes. (C) Gene Ontology enrichment analyses of the putative methylation-regulated dehydration stress memory genes. (D) Validation of the putative DNA methylation-regulated expression of stress memory genes in 5-azaC and SAM treated *B. hygrometrica* plants using qPCR. Each color cell corresponds to the log2 ratio of the relative expression value of the gene between treatment and control groups. Significant expression changes (at the 0.05 level) of genes in 5-azaC/SAM treated plants compared with the untreated control group are indicated by asterisks (*) in the corresponding squares. (E) Determination of the methylation status in promoter regions of two stress memory-related MRGs in the control and SAM-treated plants by BSP. Nucleotides are numbered relative to the putative transcription start site (+1). Arrows below each sequencing graph indicate differentially methylated sites between samples.

content, photosynthesis reduction or inhibition, mechanical damage, protein denaturation, and changes in membrane flexibility following ion leakage [4,32]. Plants resist drought stress through drought escape (rapid development and reproduction water depletion), drought avoidance (enhanced water uptake or reduced water loss to prevent dehydration), and drought tolerance (osmotic adjustment, antioxidant capacity, and desiccation tolerance) strategies [33,34]. Drought recovery, mainly refers to the plant's ability to resume growth and gain yield after exposure to severe drought stress, which causes the complete cessation of growth and loss of turgor and leaf desiccation, is regarded as another key determinant of plant drought

adaptation [4,34,35]. In the present study, the drought adaptive capability of the resurrection plant *B. hygrometrica* was estimated based on the combined phenotype and three physiological parameters, including leaf water RWC, maximal efficiency of PSII photochemistry (*Fv*/*Fm*), and electrolyte leakage (REC) upon dehydration stress and rehydration (Fig 1B). Resurrection plants retain photosynthetic activity during mild drought but regulate their metabolism by shutting down photosynthesis to minimize reactive oxygen species (ROS) accumulation during severe desiccation [10]. While poikilochlorophyllous resurrection plants disassemble their chlorophyll and degrade their thylakoid membranes to avoid photooxidative stress during desiccation, other resurrection species such as *B. hygrometrica* retain their photosynthetic apparatus during desiccation, which enables them to recover within a short period after dehydration [10,27,36,37]. The extent of electrolyte leakage as an indicator of cell membrane stability, evaluating the extent of member damage, is affected by the speed at which desiccation occurs and depends on plant species [38,39]. The significantly increased electrolyte leakage observed in both acclimated and non-acclimated plants subjected to severe dehydration might be considered as a reversible modification in the structure of the cell wall and plasma membrane caused by mechanical and oxidative stress [38]. The recovery of electrolyte leakage to control values coincides with the *Fv*/*Fm* and morphological changes in drought-acclimated *B. hygrometrica* plants after rehydration, suggesting a repair-based strategy of rapid desiccation-tolerant plants to recover from the damage incurred upon slow and rapid dehydration.

Many plant species can acclimate to drought (dehydration) stress, which is followed by a period of stress memory [11]. Enhance drought (desiccation) tolerance of acclimated plants is associated with developmental and physiological alternations such as the accumulation of compatible solutes (soluble sugars, proline, and betaine) and osmo-protective proteins (LEA proteins, ELIPs, and sHSPs), response of phytohormone (ABA and jasmonate) signaling, reduction of the photosynthetic apparatus, and induction of members of the antioxidant system to limit ROS accumulation and membrane damage [40,41]. Our previous transcriptome analysis of *B. hygrometrica* plants under drought stress had revealed the high-level accumulation of a subset of genes encoding HSP70 and sHSP that may function in protein folding in plants after drought acclimation [28]. The present study revealed that the transcription of a gene encoding HSP20 was significantly induced after drought acclimation and exhibited memory expression patterns in rehydrated plants (Fig 2B), which was consistent with the maintenance and erasing of RDT in *B. hygrometrica* plants. Genome-wide screening of drought/dehydration stress-responsive genes has indicated the establishment of drought/dehydration stress memory in many plant species, such as *Arabidopsis thaliana*, maize (*Zea mays*), and rice (*Oryza sativa*); it revealed the existence of various response patterns of memory transcripts, allowing plants to finely tune their responses to ongoing or recurring stress [19–22,25]. In this study, we constructed a novel system involving drought acclimation-induced RDT acquisition, maintenance, and erasing in *B. hygrometrica*, using non-acclimated fresh and dehydrated plants as controls (Fig 1A). Among the specific rapid dehydration-responsive genes in rapid desiccation-tolerant plants (Fig 2C and 2D), many are involved in the biosynthesis of zeatin-type cytokinins (CTKs) and phospholipids, metabolism of nitrogen and carbon, and epidermal morphogenesis (cell differentiation and lateral root development) (S2 Table). The relationship between nitrate and CTKs has long been recognized, in which nitrate-specific signal up-regulates *isopentenyltransferase 3* (*IPT3*) expression for CTK synthesis; the induction of high-affinity nitrate transporters is possibly mediated by nitrate and CTK signals [42,43]. CTK-mediated signaling is also related to the regulation of plant development, such as stimulation of defensive epidermal differentiation and suppression of lateral root development; and a wide variety of metabolic processes, including phytohormones (CTK, auxin, and gibberellin), carbon, nitrogen, and trehalose metabolism in response to drought stress [42,44,45]. Thus, it may be

speculated that the increased synthesis of phosphatidylserine, which is involved in membrane stabilization and stress signaling [46], together with the interaction of nitrate and CTK signals in the regulation of metabolism and development is a critical driving mechanism that facilitates the adaptation of rapid dehydration stress for rapid desiccation-tolerant plants. Three subsets of acclimation pre-induced rapid dehydration-responsive genes that were memorized for short-term (only in AD/A), mid-term (AD/A and A4D/A4), and long-term (AD/A, A4D/ A4, and A18D/A18) were identified as candidate genes putatively associated with the dehydration stress memory process in *B. hygrometrica* plants (Fig 2C and 2D). This result corroborates the transcriptional memory previously demonstrated in *Arabidopsis*, maize, rice, and coffee (*Coffea canephora*), plants subjected to multiple dehydration stress/recovery treatments [19– 26]. Interestingly, the transcriptome data showed that more than half of the short-term, mid-term, and long-term transcriptional memory genes upon rapid dehydration stress are possibly transmitted from the first slow dehydration event (Fig 2C-2E), demonstrating the drought acclimation-dependent establishment, maintenance, and erasing of dehydration stress memory in *B. hygrometrica* plants.

Cytosine DNA methylation functions in several processes in plant developmental and adaptive responses to the environment by modulating transposon silencing, gene expression, and chromosome interactions [47,48]. The extent and pattern of genomic DNA methylation in plants is the outcome of dynamic regulation of establishment, maintenance of methylation, and active demethylation, which are mediated by the enzymatic activities of various DNA methyltransferases and demethylases that are targeted to specific genomic regions by distinct pathways [48]. Transcriptional alternations of methylation-related genes encoding DNA methyltransferases and demethylases have been previously detected during development and drought stress in plants [31,49]. In line with previous studies, our observations showed that dynamic expression changes of DNA methylation pathway genes occurred in *B. hygrometrica* plants during drought acclimation, growth, and rapid dehydration response (Fig 3A), thus, implying the putative dynamic DNA methylation reprogramming during plant growth as well as changes in response to dehydration stress. WGBS analysis of the methylome of *B. hygrometrica* revealed much higher DNA methylation levels in CG, CHG, and CHH contexts than those in *A. thaliana* (24%, 6.7%, and 1.7% for CG, CHG, and CHH, respectively) and many crop species, such as rice (44.5%, 24.1%, and 4.7%), soybean (*Glycine max*; 63%, 44%, and 5.9%), wheat (*Triticum aestivum*; 44.5%, 24.1%, and 4.7%), and cassava (*Manihot esculenta*; 58.7%, 39.5%, and 3.5%), and similar methylation levels of CG and CHG compared to maize (86.4% and 70.9%) and spruce (*Picea abies*; 74.7% and 69.1%) (Fig 3B) [50–56]. This finding suggests that DNA methylation is more robustly maintained in *B. hygrometrica* than in most previously studied plant species, perhaps because of its large genome size (~1,691 Mb), high level of repetitive sequences (75.75%), and high GC content (42.30%), which are positively correlated with genome-wide DNA methylation levels [30,56–58]. As expected, we found slow and rapid dehydration-induced hypomethylation in all three cytosine contexts of both acclimated and non-acclimated plants (Fig 3B), consistent with previous studies demonstrating that drought and other environmental stress, such as cold, high-salinity, and heavy metals, tend to induce demethylation of genomic DNA [59–62]. In contrast, drought acclimation-induced and growth-dependent hypermethylation occurred specifically in the CHH context, which might be related to the involvement of different mechanisms maintaining DNA methylation between symmetric and asymmetric sites. In plants, symmetrical methylation at CG and CHG sites can be maintained independently of siRNAs by MET1, which is an ortholog of mammalian DNA methyltransferase DNMT1, and plant-specific CMT3, respectively [47]. Asymmetric CHH methylation, however, is maintained *de novo* through domains rearranged methyltransferase 2 (DRM2), an ortholog of the mammalian DNA methyltransferase

DNMT3A/b, via the RdDM pathway [63,64]. The transcriptional changes of the gene encoding DRM2/DNMT3A (G001986), which was observed to be significantly positively correlated with CHH methylation dynamics (Fig 3C), may contribute to the maintenance of CHH methylation in *B. hygrometrica* during plant growth and dehydration responses. Notably, sharply reduced DNA methylation levels at symmetrical sites but not at non-symmetrical sites were observed in drought-acclimated plants at an early stage of RDT maintenance (Fig 3B), although no significant differences in the transcription of methylation pathway genes were detected in the comparisons of A4/A or A4/F4 (Fig 3A). The hypomethylation status could result from self-repair mechanisms of the physiological recovery from severe dehydration stress that requires the transcriptional activation of substantially more genes [20], as evidenced by significantly elevated leaf REC in A4 plants compared with other fresh plants (Fig 1D). In addition, there was a much lower level of CHH methylation compared with that of CG and CHG in leaves of both fresh and dehydrated plants (Fig 3B), corroborating observations from a previous study on *B. hygrometrica* explants during regeneration and from other plants such as *A. thaliana*, tomato (*Solanum lycopersicum*), and maize [31,50,65,66]. However, a larger number of DMRs in the CHH context than that in CG and CHG contexts was detected between acclimated and non-acclimated plants and in plants responding to dehydration stress (Fig 3D); this may be due to the redundancy of the asymmetric CHH motif sequence and less stable of the methylation at that site [67].

Gene-associated DNA methylation in the promoter regions usually represses gene transcription; this happens directly by inhibiting the binding of transcription activators or promoting the binding of transcription repressors, or indirectly by promoting repressive histone modifications, such as H3K9me2, or inhibiting permissive histone modifications, such as histone acetylation [48]. In contrast, the gene body methylation can prevent aberrant transcription initiation from internal cryptic promoters and increase pre-mRNA splicing efficiency, but its biological function in regulating gene expression varies among different plant species [48,68,69]. Our investigations demonstrated that there was no significant correlation between transcriptional changes and methylation differences of entire genes (Figs 5 and S8–S10), even though the overall DNA methylation level in the promoter or within the gene body was associated with the suppression of gene expression in both fresh and dehydrated *B. hygrometrica* plants (Fig 4). These observations are consistent with recent genome-wide DNA methylation studies in several plants uncovering various associations between methylation variations and gene expression differences and suggest that promoter methylation plays an indecisive role in transcriptional regulation during the processes of RDT acquisition, maintenance, and erasing [31,49,70,71]. Nevertheless, there is evidence for the possible control of DNA methylation for a wide variety of genes differentially expressed in plants during RDT acquisition, maintenance, and erasing (Figs 5 and S11), according to the negative correlation between the transcriptional changes and the relative alternation of the promoter methylation [48]. Interestingly, MRGs involved in genetic information processing were significantly enriched, e.g., the process of DNA conformation, RNA splicing, translation, and post-translational protein modification (Fig 6 and S5–S8 Tables), which suggests that DNA methylation may extensively participate in the fine-tuning of fundamental processes during plant acclimation, growth, and rapid dehydration stress response. Significant enrichment of rapid dehydration stress-responsive MRGs was also observed involving plant purine metabolism, one of the fundamental steps in nitrogen recycling required for sustaining normal plant growth and development as well as plant response and adaptation to environmental stress [72,73]. These results are also consistent with previous findings that DNA methylation contributes to regulating fundamental processes underpinning cell differentiation, growth, and stress response [31,49,67,74]. Intriguingly, we detected several putative methylation-regulated short- and mid-term dehydration stress

memory genes (Fig 7A and 7B), and demonstrated the direct or indirect regulation of promoter methylation/demethylation on their transcription activity (Fig 7D), further supporting the potential link between DNA methylation and plant dehydration response and memory in *B. hygrometrica* plants. Among them, two putative methylation-regulated mid-term dehydration stress memory genes encoding PSF38A and USPase, which were up-regulated in rapid desiccation-tolerant plants subjected to dehydration stress (S12 Fig), have been shown to be involved in plant development and stress responses [75,76]. Previous studies have demonstrated the wide occurrence of splicing factor-mediated alternative splicing events at particular developmental stages or in response to environmental changes, particularly in ABA-mediated stress responses [75]. USPase is the major enzyme responsible for the synthesis of raffinose family oligosaccharides, which can protect plant cells during seed desiccation and drought stress [76]. Moreover, *B. hygrometrica AVT1L* encodes a homolog of yeast AVT, which is also hypomethylated and up-regulated in rapid desiccation-tolerant plants subjected to dehydration stress (S12 Fig). This homolog is reportedly involved in the transport of large neutral amino acids including glutamine (asparagine), isoleucine (leucine), and tyrosine into vacuoles, thus maintaining the homeostasis of vacuolar and cytosolic amino acids [77]. Therefore, we hypothesized that drought stress-induced hypomethylation and upregulation of memory genes associated with alternative splicing changes and accumulation of oligosaccharides and vacuolar amino acids, allow rapid adjustment of the abundance and function of key stress-response components, which ultimately contribute to the improvement of dehydration tolerance in *B. hygrometrica*. In this context, upcoming studies will aim to characterize the biological importance of stress memory genes and the underlying molecular mechanisms for epigenetic regulation of their expression during plant stress responses.

## Materials and methods

### Plant materials and treatments

Seeds of *B. hygrometrica* were collected from a naturally growing population of approximately 200 wild plants inhabiting the Western Hills of Beijing, China (40˚12 N, 116˚10 E). *B. hygrometrica* is a mixed-mating species and its mating system is mainly outcrossing; therefore, the collected seeds are possibly from both genetically independent parents and sib/half-sib families in the population [31]. All the collected seeds were mixed in one tube and sown in ordinary potting soil after vernalization at 4˚C for 3 d. The seedlings were grown under well-watered conditions in a greenhouse with a photoperiod of 15 h light/9 h dark, 25˚C, and 60% relative humidity (RH). For drought acclimation treatment, three-month-old seedlings were subjected to slow-onset water deprivation treatment by withholding water for 14 d (slow soil-drying) after saturating soil moisture, and then re-watered for 3 d [28,29]. Untreated plants served as the non-acclimated control group. Rapid dehydration (rapid air-drying) treatment was imposed by transferring plants from soil to Petri dishes and drying immediately at 25˚C and 60% RH for 2 d. The air-dried plants were rehydrated by subjecting them to wet filter paper in Petri dishes for 3 d. Both acclimated and non-acclimated plants were subjected to rapid air-drying and rehydration after continuous cultured for 0, 4, 8, 13, 18, and 22 w, respectively (Fig 1A). For SAM and 5-azaC treatments, surface-sterilized seeds were vernalized and germinated on half-strength Murashige-Skoog (1/2 MS) medium according to the method described by Sun *et al.* [31]. After 45 d of culture, the seedlings were transferred to 1/2 MS medium supplemented with either SAM (0, 10, 50, and 100 μM) or 5-azaC (0, 10, 50, and 100 μM) and continued to grow for 7 d. Each treatment was applied to at least 30 individual plants, with six biological replicates.

## Physiological measurements

The RWC of *B. hygrometrica* leaves was determined according to Lin *et al.* [78] with minor modifications, where all of the leaves cut from each plant were weighed immediately to record fresh weight (FW). Subsequently, the detached leaves were immersed in distilled water at room temperature for 6 h, weighed to record turgid weight (TW), and subjected to fast oven-drying at 105˚C for 1 h followed by 65˚C for several hours to a stable weight to record dry weight (DW). The RWC was calculated as follows: RWC (%) = (FW − DW)/(TW − DW) × 100. The chlorophyll fluorescence parameter *Fv/Fm* was monitored in intact leaves after dark adaptation for 1 h using the Maxi-Imaging-PAM (Walz, Germany), with a saturating light intensity of approximately 800 mmol $m^{-2}$ $s^{-1}$ and duration of 4.5 s. Leaf REC was measured using an EC 125 Conductivity Meter (Hanna Instruments, Padova, Italy), following the method described previously [79]. Six individual plants from each biological replicate were used to measure the physiological indexes.

## RNA-seq and data analysis

Approximately 45 leaves detached from at least 15 individual *B. hygrometrica* plants for each biological replicate were pooled for transcriptome profiling analysis. Three biological replicates were performed for each treatment group. Total RNA was extracted using a Plant Total RNA Extraction kit (Sigma, St. Louis, MO, USA) and purified using DNase I (Promega, Madison, WI, USA). The cDNA synthesized from enriched poly(A) mRNA was connected with adaptors, PCR-amplified, and then sequenced using an Illumina Hi-Seq 4000 platform (Illumina Inc., San Diego, CA, USA), with 150-bp paired-end reads (PE150). Filtered sequencing reads were aligned against the reference genome of *B. hygrometrica using* HISAT [80]. Novel transcripts were predicted after transcript reconstruction and comparison with reference transcripts using StringTie and Cuffcompare, respectively [81,82]. To calculate the transcript abundance of each gene, clean reads were mapped to the reference using Bowtie followed by estimating the gene and isoform expression levels with the fragments per kilobase per million fragments mapped (FPKM) method using RSEM [83,84]. DEGs between samples were detected using DEseq2, with fold change ≥ 2 and adjusted $p \leq 0.05$ as cutoffs for statistically significant differences in expression [85]. GO and KEGG enrichment analyses of DEGs between samples were performed using the OmicShare online platform (http://www.omicshare.com/tools/). The GO terms/KEGG pathways with a corrected *p*-value (Q-value) ≤ 0.05 were considered significantly enriched.

## WGBS and data analysis

Genomic DNA used for WGBS was extracted from the leaves of *B. hygrometrica* using a Plant Genomic DNA Kit (Tiangen Biotech, Beijing, China). Approximately 45 leaves detached from at least 15 individual plants were pooled for DNA extraction in each biological replicate. The WGBS library of each treatment was constructed from a pool of equal quantities of DNA samples from six biological replicates. Purified genomic DNA was subjected to sonication, 3'A addition, adaptor ligation, followed by bisulfite modification with a ZYMO EZ DNA methylation-Gold kit (Zymo Research, Orange, CA, USA), gel purification, and PCR amplification. Sequencing was performed on an Illumina HiSeq X Ten platform (Illumina Inc., San Diego, CA, USA) with PE150 mode. Raw sequencing reads were filtered by removing the adaptors and discarding low-quality and N-base-containing reads. Clean reads were aligned to both the positive and negative strands of the reference genome using BSMAP [86], allowing up to two mismatches. The DNA methylation level of 5-methylcytosine (m5C) was calculated as follows: methylation level (%) = methylated reads/(methylated reads + unmethylated reads) × 100.

Differentially methylated cytosines were determined using Fisher's exact test. Changes in methylation of at least 20% and $p < 0.05$, were considered significant. Identification of DMRs between samples was performed according to previously described methods [31,87]. In brief, the first five adjacent CG/CHG/CHH sites containing at least four CG/CHG/CHH sites with the same changing trend and a Wilcoxon rank-sum test $p < 0.05$, were considered as candidate DMRs. The 3' downstream adjacent CG/CHG/CHH sites with the same changing trend were incorporated with the candidate DMRs until the Wilcoxon rank-sum test $p \geq 0.05$. An inter-distance of 200-bp was allowed between the two adjacent CG/CHG/CHH sites. GO enrichment analysis of MRGs between samples was performed as described above.

## Quantitative real-time PCR

The total RNA of *B. hygrometrica* plants used for qPCR was extracted and purified using the method described above. Preparation of cDNA and qPCR was performed according to Sun *et al.* [88]. Gene expression levels are presented as the mean ± SD by the Ct method using *elongation factor 1α* (*EF1α*) and *18S* rRNA as the reference genes [31]. The gene-specific primers used for qPCR are listed in S11 Table. Three biological replicates were analyzed using the mean values of the three technical replicates for each sample.

## Bisulfite sequencing PCR

The genomic DNA of SAM/5-azaC treated *B. hygrometrica* seedlings used for BSP was extracted as described above. Bisulfite treatment of DNA was performed using a DNA bisulfite conversion kit (Tiangen Biotech, Beijing, China). Each targeted DMR was amplified from the bisulfite-converted DNA with AceTaq DNA Polymerase (Vazyme, Nanjing, China) and degenerate primers designed using the Methyl Primer Express v1.0 software (see S11 Table for sequences details), followed by Sanger sequencing to determine its methylation status.

## Statistical analysis

The statistical significance of the physiological indexes among samples and gene expression between the control and SAM/5-azaC treatment groups were determined by one-way ANOVA followed by Tukey's HSD test and by an independent sample t-test using SPSS 20.0 for windows (SPSS Inc., Chicago, IL, USA), respectively. Correlation analysis between DNA methylation and the expression of methylation pathway genes was performed using SangerBox software (http://sangerbox.com/). Venn diagrams were constructed using the online tool jvenn (http://jvenn.toulouse.inra.fr/app/example.html/). HCA and PCA were performed using MetaboAnalyst 3.0 (http://www.metaboanalyst.ca/).

## Supporting information

**S1 Fig. Hierarchical clustering analysis (HCA) and principal component analysis (PCA) of the transcriptome data.** (A) HCA dendrogram showing the discrimination among samples based on the transcriptome data. (B) Score scatter plot of PCA showing the discrimination among samples based on the transcriptome data. The first principal component (PC1) and the second principal component (PC2) explained 67.7% and 11.6% of the variation in the initial data sets, respectively.
(TIF)

**S2 Fig.** Venn diagrams displaying the overlap of up- (A) and down-regulated (B) DEGs between the acclimated and non-acclimated fresh plants at three growth stages.
(TIF)

**S3 Fig.** Venn diagrams displaying the overlap of up- (A) and down-regulated (B) DEGs during plant growth.
(TIF)

**S4 Fig.** Venn diagrams displaying the overlap of up- (A, C, E) and down-regulated (B, D, F) DEGs between acclimated and non-acclimated fresh plants responding to rapid dehydration stress at each of the three growth stages.
(TIF)

**S5 Fig. Hierarchical clustering heatmap showing the expression changes of RDT-specific DEGs in fresh plants responding to rapid dehydration stress.** RDTP, rapid dehydration-tolerant plants; RDSP, rapid dehydration-sensitive plants. Scale bar represents log2 fold-change expression (red, upregulation; blue, downregulation) between samples.
(TIF)

**S6 Fig. Hierarchical clustering heatmap showing the expression changes of short-term transcriptional memory genes in plants during acclimation (SD/F and A/F) and responding to rapid dehydration stress (FD/F, AD/A, F4D/F4, A4D/A4, F18D/F18, and A18D/A18).** Acc, the acclimation process; RDTP, rapid dehydration-tolerant plants; RDSP, rapid dehydration-sensitive plants. Scale bar represents log2 fold-change expression (red, upregulation; blue, downregulation) between samples.
(TIF)

**S7 Fig. Hierarchical clustering heatmap showing the expression changes of long-term transcriptional memory genes in plants during acclimation (SD/F and A/F) and responding to rapid dehydration stress (FD/F, AD/A, F4D/F4, A4D/A4, F18D/F18, and A18D/A18).** Acc, the acclimation process; RDTP, rapid dehydration-tolerant plants; RDSP, rapid dehydration-sensitive plants. Scale bar represents log2 fold-change expression (red, upregulation; blue, downregulation) between samples.
(TIF)

**S8 Fig. Comparison of the DNA methylation level of DEGs between samples.** Green and blue lines represent the methylation levels of up-regulated DEGs in the former and latter samples of each comparison, respectively. Red and black lines represent the methylation levels of down-regulated DEGs in the former and latter samples of each comparison, respectively. U2k, 2-kb sequences upstream of the protein-coding region; D2k, 2-kb sequences downstream of the protein-coding region.
(TIF)

**S9 Fig. Correlation analysis of gene expression differences and promoter methylation variations in the CHH contexts between samples.** DEGs and non-DEGs are indicated by red and blue dots, respectively. Cor, Pearson correlation coefficient.
(TIF)

**S10 Fig. Correlation analysis of CG-, CHG-, and CHH-type of DMRs within the gene promoter regions and gene expression differences between samples.** DEGs with CG-, CHG-, and CHH-type of DMRs in their promoter regions are indicated by red, blue, and magenta dots, respectively. Non-DEGs with CG-, CHG-, and CHH-type of DMRs in their promoter regions are indicated by yellow, green, and grey dots, respectively. Cor, Pearson correlation coefficient.
(TIF)

**S11 Fig. Venn diagrams displaying the overlap of MRGs among comparisons.** (A) Putative MRGs in plants subjected to slow dehydration stress (SD/F) and rehydration (A/SD). (B) Putative MRGs between acclimated (A, A4, and A18) and non-acclimated (F, F4, and F18) fresh plants. (C) Putative MRGs during plant growth. (D) Putative MRGs in both acclimated and non-acclimated fresh plants subjected to rapid dehydration stress.
(TIF)

**S12 Fig.** Heatmap showing the expression changes of putative methylation-regulated mid-term dehydration stress memory genes in fresh plants subjected to dehydration stress. Acc, plants subjected to drought acclimation; RDT and RDS represent rapid desiccation-tolerant and -sensitive plants subjected to dehydration stress, respectively. Scale bar represents log2 fold-change expression (yellow, upregulation; green, downregulation) between samples. Genes with significant expression differences (fold change $\geq$ 2 and adjust $p \leq 0.05$) are indicated by asterisks (*) in the squares.
(TIF)

**S1 Table Summary of the transcriptome sequencing data.**
(XLSX)

**S2 Table GO and KEGG enrichment analyses of the specific rapid dehydration-responsive genes in rapid desiccation-tolerant plants.**
(XLSX)

**S3 Table Summary of the WGBS data.**
(XLSX)

**S4 Table The average methylation level of the gene promoter regions of all samples.**
(XLSX)

**S5 Table GO enrichment analysis of acclimation-related MRGs.**
(XLSX)

**S6 Table GO enrichment analysis of MRGs between acclimated and non-acclimated fresh plants.**
(XLSX)

**S7 Table GO enrichment analysis of MRGs during growth.**
(XLSX)

**S8 Table GO enrichment analysis of MRGs during responding to rapid dehydration stress.**
(XLSX)

**S9 Table List of the putative methylation-regulated short-term and mid-term dehydration stress memory genes.**
(XLSX)

**S10 Table GO enrichment analysis of the putative methylation-regulated dehydration stress memory genes.**
(XLSX)

**S11 Table List of the primers used in this study.**
(XLSX)

**S1 Dataset Data used to make all the figures and statistical analyses.**
(XLSX)

## Acknowledgments

The authors kindly acknowledge Dr. Chih-Ta Lin, Jin-Ying Ma, and Tong Zhao for their assistance in the preparation of plant materials. The authors would like to thank Editage (www.editage.cn) for English language editing. The authors are also grateful to the Beijing Genomics Institute (Beijing, China) and the Hengchuang Gene Technology Co. Ltd. (Shenzhen, China) for technical assistance.

## Author Contributions

**Conceptualization:** Run-Ze Sun, Jie Liu, Yuan-Yuan Wang, Xin Deng.

**Data curation:** Run-Ze Sun.

**Formal analysis:** Run-Ze Sun.

**Funding acquisition:** Run-Ze Sun, Xin Deng.

**Investigation:** Run-Ze Sun, Jie Liu, Yuan-Yuan Wang.

**Methodology:** Run-Ze Sun, Jie Liu, Yuan-Yuan Wang, Xin Deng.

**Project administration:** Xin Deng.

**Resources:** Run-Ze Sun, Jie Liu, Yuan-Yuan Wang.

**Software:** Run-Ze Sun.

**Supervision:** Xin Deng.

**Validation:** Run-Ze Sun, Jie Liu, Yuan-Yuan Wang.

**Visualization:** Run-Ze Sun.

**Writing – original draft:** Run-Ze Sun.

**Writing – review & editing:** Run-Ze Sun, Jie Liu, Yuan-Yuan Wang, Xin Deng.

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
