## [Decision Letter · Decision Letter 0]

10 Feb 2021

Dear Dr Deng,

Thank you very much for submitting your Research Article entitled 'DNA methylation-mediated modulation of priming and memory of rapid dehydration tolerance in the resurrection plant Boea hygrometrica' to PLOS Genetics.

The manuscript was fully evaluated at the editorial level and by three independent peer reviewers. The reviewers appreciated the attention to an important problem, but raised some substantial concerns about the current manuscript. Based on the reviews, we will not be able to accept this version of the manuscript, but we would be willing to review a much-revised version. We cannot, of course, promise publication at that time.

If you decide to revise the manuscript for further consideration at PLOS Genetics, please aim to resubmit within the next 60 days, unless it will take extra time to address the concerns of the reviewers, in which case we would appreciate an expected resubmission date by email to plosgenetics@plos.org.

[LINK]

We are sorry that we cannot be more positive about your manuscript at this stage. Please do not hesitate to contact us if you have any concerns or questions.

Yours sincerely,

Ian Henderson

Associate Editor

PLOS Genetics

Li-Jia Qu

Section Editor: Plant Genetics

PLOS Genetics

Reviewer's Responses to Questions

**Comments to the Authors:**

Reviewer #1: The manuscript by Sun et al., is primarily aimed to understand the molecular mechanisms underlying dehydration tolerance, more specifically the mechanism by which drought stress memory is attained in the “resurrection plant” Boea hygrometrica. The study starts with an interesting physiologic experiment that proves that the “memory” of drought stress exists in Boea hygrometrica, lasting up to 18 weeks.

Then the authors continue the work with the transcriptomic analysis and the profiling of DNA methylation at several steps during acclimation, recovery… but it seems like in their effort to be exhaustive, they describe every figure precisely with no apparent direction nor focus in what they considered important or more specific. This study provides a formidable collection of raw data that has the potential of yielding an exciting output, but one finish the paper with the feeling of these data being under-investigated.

Reviewer #2: The manuscript by Sun et al., studied the mechanism of acclimation-induced stress memory in Boea hygrometrica by global transcriptome analysis and DNA methylation comparison. It’s very interesting for understanding the resistance mechanism of botany under global climate change. Here we can found that a large number of omics analyses and investigation for uncovering this question have been done. The results and conclusion are reliable and solid, but there are some questions need to be addressed.

Major comments:

- Fig. 3B, the mCG and mCHG level in "A4" is markedly lower than those of other samples. This might be due to the available of only a single biological replicate of WGBS, or the sample-specific expression patterns of the methylation enzymes? The authors should discuss this aspect.

- line 375, what is the relationship between CHH methylation differences and gene expression changes? I think the authors should analyse and discuss about this, since "larger number of DMRs in CHH context than in CG and CHG contexts was detected" (line 561).

- line 621, more information about the plant sampling are needed, e.g., maternal genotypes, populations, outcrossed or self-pollinated? Are the collected seeds from genetically independent parents or sib families? All of these information must be addressed.

- A brief explanation about how did the drought acclimation be done and how long did the dehydration/rehydration treatment last should be mentioned in the Materials and Methods, although references were cited.

Minor comments:

- Line 23, incomplete presented.

- Line 155, the result of statistical analysis (P-value?) is missing. It would be important clarify "significant" in the text.

- Line 203, differentiate between "rapid desiccation-tolerance (RDT)" and "rapid dehydration tolerance (RDT)".

- Fig. 1, what do the error bars represent in the plots?

- Figs. 3A, 6D, what do the asterisks represent in the heatmaps? Not clear.

- Supplemental figure and table legends are not self-explanatory. More details should be given.

Reviewer #3: 1. The authors did the qPCR of 5-azaC/SAM treated B. hygrometrica seedlings and BS-PCR for addressing the role of DNA methylation with RDT. However, they didn't validate the results from transcriptomes by qPCR or any other method.

2. The language aspect needs to be refined.

**Have all data underlying the figures and results presented in the manuscript been provided?**

Reviewer #1: Yes

Reviewer #2: Yes

Reviewer #3: Yes

PLOS authors have the option to publish the peer review history of their article (what does this mean?). If published, this will include your full peer review and any attached files.

Reviewer #1: **Yes: **Israel Ausin

Reviewer #2: No

Reviewer #3: No

---

## [Editor Report · Decision Letter 1]

14 Apr 2021

Dear Dr Deng,

We are pleased to inform you that your manuscript entitled "DNA methylation-mediated modulation of rapid desiccation tolerance acquisition and dehydration stress memory in the resurrection plant Boea hygrometrica" has been editorially accepted for publication in PLOS Genetics. Congratulations!

Yours sincerely,

Ian Henderson

Associate Editor

PLOS Genetics

Li-Jia Qu

Section Editor: Plant Genetics

PLOS Genetics

Comments from the reviewers (if applicable):

**Data Deposition**

http://datadryad.org/submit?journalID=pgenetics&manu=PGENETICS-D-20-01850R1

**Press Queries**

---

## [Editor Report · Acceptance letter]

27 Apr 2021

PGENETICS-D-20-01850R1 

DNA methylation-mediated modulation of rapid desiccation tolerance acquisition and dehydration stress memory in the resurrection plant Boea hygrometrica 

Dear Dr Deng, 

We are pleased to inform you that your manuscript entitled "DNA methylation-mediated modulation of rapid desiccation tolerance acquisition and dehydration stress memory in the resurrection plant Boea hygrometrica" has been formally accepted for publication in PLOS Genetics! Your manuscript is now with our production department and you will be notified of the publication date in due course.

With kind regards,

Katalin Szabo

PLOS Genetics

On behalf of:
